# Nanocellulose-Reinforced Polyurethane for Waterborne Wood Coating

**DOI:** 10.3390/molecules24173151

**Published:** 2019-08-29

**Authors:** Linglong Kong, Dandan Xu, Zaixin He, Fengqiang Wang, Shihan Gui, Jilong Fan, Xiya Pan, Xiaohan Dai, Xiaoying Dong, Baoxuan Liu, Yongfeng Li

**Affiliations:** 1State Forestry and Grassland Administration Key Laboratory of Silviculture in Downstream Areas of the Yellow River, Shandong Agricultural University, No. 61 Daizong Road, Taian 271018, China; 2Key Laboratory of Bio-based Material Science and Technology of Ministry of Education, Northeast Forestry University, Harbin 150040, China; 3Shandong Laucork Develepment Co. Ltd., Room 401, building A2, High-tech Zone, Jining 272100, China; 4Qingdao Institute of Biomass Energy and Bioprocess Technology, Chinese Academy of Sciences, No.189 Songling Road, Qingdao 266101, China

**Keywords:** waterborne polyurethane, nanocellulose, wood coatings, enhancement

## Abstract

With the enhancement of people’s environmental awareness, waterborne polyurethane (PU) paint—with its advantages of low release of volatile organic compounds (VOCs), low temperature flexibility, acid and alkali resistance, excellent solvent resistance and superior weather resistance—has made its application for wood furniture favored by the industry. However, due to its lower solid content and weak intermolecular force, the mechanical properties of waterborne PU paint are normally less than those of the traditional solvent-based polyurethane paint, which has become the key bottleneck restricting its wide applications. To this end, this study explores nanocellulose derived from biomass resources by the 2,2,6,6-tetramethylpiperidine-1-oxyl (TEMPO) oxidation method to reinforce and thus improve the mechanical properties of waterborne PU paint. Two methods of adding nanocellulose to waterborne PU—chemical addition and physical blending—are explored. Results show that, compared to the physical blending method, the chemical grafting method at 0.1 wt% nanocellulose addition results in the maximum improvement of the comprehensive properties of the PU coating. With this method, the tensile strength, elongation at break, hardness and abrasion resistance of the waterborne PU paint increase by up to 58.7%, ~55%, 6.9% and 3.45%, respectively, compared to the control PU; while the glossiness and surface drying time were hardly affected. Such exploration provides an effective way for wide applications of water PU in the wood industry and nanocellulose in waterborne wood coating.

## 1. Introduction

As the surface layer of objects, coatings are widely used in the daily civil life of furniture, packaging, construction, automobiles and other military fields such as ships and weapons. They can not only provide protection for materials, resisting external factors such as environment, microorganisms and man-made damage, but also provide a decorative effect for materials, giving materials a beautiful color and glossiness [1]. However, traditional coatings are mostly solvent-based and oil-based, which release volatile organic compounds (VOCs) and thus pollute the environment. Against the background of increasing environmental awareness of human beings and the requirement to strengthen environmental protection throughout the world, waterborne coating with water as the solvent or carrier, which releases hardly any VOCs, is regarded as one of the ideal environmental protection coatings to replace traditional organic solvent-based paint and is favored by the industry [2,3]. Wood furniture is one of the main fields of paint application. Waterborne polyurethane (WPU) is one of the most popular waterborne coatings for wood. In addition to a lower release of volatile organic compounds and lower odor, it also has higher peel strength, lower temperature flexibility, glossiness, chemical resistance, easy cleanability, low viscosity and weather resistance and other advantages. However, the mechanical strength of the film is much lower than that of traditional solvent-based coatings, which has become a key bottleneck affecting the wider application of waterborne polyurethane paints in the wood field [4,5].

Nanomaterials have been widely used to improve the properties of waterborne polyurethane (PU) due to their small size effect and surface effect, such as nano-silica, nano-zinc oxide, nano-titanium dioxide, clay, cellulose nanocrystals, and so forth [6,7,8,9,10,11,12,13,14,15]. The mechanical properties, corrosion resistance and barrier properties of waterborne polyurethane have also been improved to varying degrees. However, there are still some problems such as poor compatibility between nanomaterials and polyurethane resin, easy agglomeration and the lower aspect ratio of the nanomaterials, which affect the glossiness and mechanical strength of the paint film [16,17,18,19,20,21,22,23,24].

Nanocellulose fiber refers to cellulose nanofibers with a diameter of 1-100 nm and a length of more than 1um. It is widely derived from natural plant fibers and has the characteristics of renewable, degradable, biocompatible and so on. It also has the advantages of high mechanical strength, high aspect ratio, and being easy to disperse in water and easy to modify due to abundant surface active groups (like -OH, -COOH), which make it one of the most promising nanomaterials that has attracted much attention in recent years [25,26,27,28,29,30,31]. This study is expected to reinforce waterborne polyurethane paint film without an obvious negative impact on the glossiness by nanocellulose with ultra-fine size, high aspect ratio, rich active functional groups, and a network structure via physical or chemical addition methods (Figure 1) [32,33,34,35]. It provides a new strategy for the improvement of the properties of waterborne polyurethane and its wide application to wood lacquer. Such a strategy has the advantages of green and renewable utilization of biomass materials, environment-friendly and scalable processing of the paint.

## 2. Discussion

The biomass material was treated by purification, followed by TEMPO oxidation and high-pressure homogenization to obtain nanocellulose with a fine and uniform size [27,28,31,32]. Figure 2a shows the dispersion morphology of raw wood flour, purified cellulose and nanocellulose in water. Among them, the left one is the wood powder deposited on the bottom of the water with light brown and the middle is the white purified cellulose precipitate with transparent water. Since they all present in an aggregate state with a mean diameter over micrometer size, the two samples precipitate on the bottom of the water. The right one shows the nanocellulose/water suspension with light blue transparency, indicating that the cellulose is uniformly dispersed in water with a fine nanometer structure at a stable suspension state, causing the Rayleigh scattering phenomenon when natural light passes through the water suspension. AFM confirms the fine structure of nanocellulose with a uniform diameter of ~10nm (Figure 2b,c), which corresponds to the light blue transparency of the nanocellulose suspension in Figure 2a. Additionally, Figure 2b shows that the nanocelluloses present in the network structure in the suspension by interacting with each other, which could contribute to the reinforcement of the polymer matrix [36]. The SEM observation shows that the nanocelluloses present in a network structure with length over 10 um (Figure 2d), indicating the aspect ratio of nanocellulose larger than 1000 (Figure 2c,d). Figure 2e further confirms the fine structure of the nanocellulose with a uniform diameter at the nanometer scale, which could play an important role in the reinforcement of the polymer matrix [23,24,25,26,27,28,29].

Figure 2f shows that the relative crystallinity of wood flour, holocellulose and the purified cellulose reaches 54.63%, 64.01% and 79.67%, respectively, which should be ascribed to the fact that the amorphous lignin and hemicellulose are removed successively in the purification processes, resulting in the highest crystallinity of the purified cellulose. Although the relative crystallinity of the nanocellulose after TEMPO oxidation combined with high pressure homogenization treatments decreases to 50.97%, the value is still high and beneficial to the reinforcement of the polymer matrix. The slight decrease should be mainly attributed to the disintegration of the cellulose bundles from a micrometer to a nanometer in diameter, which destroys the aggregation of the cellulose chains and thus decreases the relative crystallinity.

Figure 2g shows the electronic photos of the control waterborne polyurethane emulsion (the left bottle) and waterborne polyurethane emulsion by physical (middle one) and chemical addition (right one) of nanocellulose with amount of 0.1 wt%. From the appearance, there is no essential difference among the three, which all show milky white. TEM observation (Figure 2h,i) further shows that regardless of the physical blending method or the chemical modification method, the nanocellulose is uniformly dispersed in the waterborne polyurethane in the network winding state. The length is much longer than 10 micrometers and the diameter is in the nanometer scale, consistent with the AFM characterization results. It indicates that the nanocellulose can be uniformly and monodispersed in the waterborne polyurethane without obvious agglomeration, which lays a foundation for further modification of the polyurethane.

Nanocellulose could theoretically reinforce the mechanical properties of the PU especially when the resin as film is stretched or bent due to the interlocking of the nanocelluloses with a high specific strength and abundant active groups to assist their compatibility (Figure 3a,d) [23,24,25]. Figure 3b,c shows that the tensile strength (TS) and tensile elongation at break (TE) of the PU film are improved to some degree by the physical addition of nanocelluloses with varying content from 0.1 wt% to 0.4 wt% (accounting for the solid content of PU). When the addition of nanocellulose is 0.1 wt%, the TS and TE of the physically-modified PU (PM PU) film reaches 36.74 MPa and 370%, which increases by 42.9% and 38.58% over the control (the pure PU film), respectively. The results should be ascribed to the good dispersion of nanocellulose, which is proved by the TEM observation in Figure 2h. While to the addition of nanocellulose with 0.2 wt% and 0.4 wt%, the improvements of TS and TE of the PM PU are lower than those of PM PU with 0.1 wt% addition, which may be due to a certain degree of aggregation of the nanocellulose with a larger addition and the existence of the critical threshold [1,2,23,24,25].

The chemically-modified PU (CM PU) presents similar results to those shown in Figure 3e,f. The TS and TE attains the largest value with an improvement of ~59% and ~55% compared to the pure PU, respectively, when adding nanocellulose with a content of 0.1 wt% and 0.2 wt%. Addition of nanocellulose with 0.4 wt% results in lower TS and TE of the CM PU, even lower than that of the control PU (Figure 3f). Compared with the physical modification, the chemical modification appears to provide a great improvement in TS and TE, as proved from the results of 0.1wt% addition. Such a phenomenon reveals that the chemical modification further improves the compatibility of the nanocellulose and PU film [23].

Glossiness reflects the surface roughness via specular reflection of the visible light against surface [37]. Figure 4a,d shows a similar appearance of the PU films with and without nanocellulose modification, despite physical or chemical addition (PM PU and CM PU), respectively, which indicates a slight effect on the surface roughness induced by the addition of nanocellulose at 0.1 wt%. Figure 4b proves that the physical addition of nanocellulose results in a slight decrease of glossiness compared to the control PU coating, despite the nanocellulose content ranging from 0.1 wt% to 0.4 wt%. However, the one with 0.1 wt% nanocellulose addition shows the slight decrease of glossiness, which indicates a relatively good dispersion of nanocellulose in the PU coating resulting in a slight roughness. Similar results are observed in Figure 4e. Compared to the PM PU coating, the CM PU coating presents a slightly higher glossiness, which should be ascribed to the better compatibility between PU and nanocellulose via chemical addition.

Figure 4c shows that the PM PU coating with 0.1 wt% nanocellulose addition presents the highest hardness among the control and the PM PU coatings with different amounts of nanocellulose addition. The CM PU coating with 0.1 wt% addition of nanocellulose appears to have a similar result with the biggest value of hardness compared to the control and other CM PU coatings (Figure 4f). Other PM PU and CM PU coatings show relatively lower values, maybe attributed to the aggregation of nanocellulose with more content [23,24,25,26,27,28,29,30]. The 6.98% and 14.05% hardness improvement for the PM PU and CM PU coating, compared to the control PU, proves that the even dispersion of the nanocellulose contributes to the improvement of the mechanical properties of the PU coating. Obviously, the CM PU presents a higher hardness due to the better dispersion of nanocellulose, compared to the PM PU coating. All the results prove that the 0.1 wt% addition of nanocellulose apparently improves the hardness and negligibly decreases the glossiness due to good dispersion of nanocellulose, especially for the CM PU coating. Such an explanation corresponds to the above improvements of tensile strength and tensile elongation at break.

The abrasion resistance of wood coating in terms of mass loss rate represents the coating durability against the friction force. Figure 5a–c show that 0.1 wt% nanocellulose addition decreases the mass loss for each PM PU or CM PU, that corresponds to an improvement of 7.4% and 3.45%, respectively, compared to the control. For the CM PU, 0.2 wt% nanocellulose addition remarkably improves the abrasion resistance of 33.33%. The improvements should be also ascribed to the good dispersion of nanocellulose resulting in tight interaction among the PU macromoleculars. Similar phenomena are observed on the surface drying time of the PU coatings (Figure 5d–f). For the PM PU, both 0.2 wt% and 0.4 wt% nanocellulose additions slightly improve the surface drying time of the coating, while 0.1 wt% nanocellulose addition just slightly prolongs the surface drying time of 3.45% without obviously negative effect, compared to the control PU coating. In addition, for the CM PU, 0.4 wt% nanocellulose additions slightly improves the surface drying time of the coating, while 0.1 wt% nanocellulose addition just slightly prolongs the surface drying time of 3.45% without obviously negative effect, compared to the control PU coating. Such results maybe explained that the surface drying time of the PU coating could not be basically affected by 0.1 wt% nanocellulose addition; while for more nanocellulose addition, the surface drying time may be relatively obviously affected due to the potential existence of a critical threshold and some degree of aggregation of nanocellulose [23,24,25,26,27].

In short, based on the above comparative data of tensile strength, elongation at break, glossiness and hardness, the chemically-modified polyurethane with a 0.1 wt% nanocellulose addition performs better than the physically-modified PU and also the control PU. Although the abrasion resistance of the CM PU with an addition of 0.2 wt% nanocellulose presents the highest value of all the other CM PU and PM PU, both the CM PU and PM PU contribute positively to the abrasion resistance when the nanocellulose addition is 0.1 wt%. For the surface drying time, both the two nanocellulose addition methods almost hardly affect the PU. Consequently, the chemical addition of 0.1 wt% nanocellulose to the PU seems to be the optimized modification method for waterborne PU as a wood coating.

Theoretically, the PU coating and the nanocellulose could form a hydrogen bond between the carbamate group and the hydroxyl/carboxyl group via a physical blending method (Figure 6a); while they could form a chemical bond from the carbamate group via the chemical reaction of the isocyanate group and the hydroxyl/carboxyl group (Figure 6d). Both the methods could improve the interfacial compatibility between the PU and nanocellulose, which accordingly improves the comprehensive properties of the PU. In addition, good dispersion of the nanocellulose could further promote the compatibility of the PU and nanocellulose.

Figure 6b shows that, compared to the FTIR curves of the PM PU and the control PU, a new peak appears at 1541 cm^−1^ of the CM PU, representing the deformation vibration of the (CO)-NH group, instead of the original peak at 2259 cm^−1^ which represents the asymmetric stretching vibration of the -NCO group. Additionally, the width of the peak of the control PU at 3300 cm^−1^, representing the stretching vibration of -N-H, becomes slightly narrow after the addition of nanocellulose despite each physical blending or chemical method, which should be ascribed to the improvement of compatibility of nanocellulose and PU by hydrogen bonding or chemical bonding. Such information indicates that the -NCO group of the isopentanedione diisocyanate successfully bonded to the hydroxyl/carboxyl group of nanocellulose and changed into the CONH group, indicating the chemical reaction between nanocellulose and the PU.

Normally, the sample crystallinity could be calculated from the X-ray spectra by the Segal method [28]. Figure 6c shows that the three curves present similar forms, indicating unchanged crystallinity of the modified PU chains, compared to the control PU. In other words, the nanocellulose addition in each method does not affect the aggregation states of the PU chains. Figure 6e,f show that the TG/DTG curves of the two modified paint films are almost the same as those of the control PU film, which indicates that the addition of nanocellulose has no significant effect on the thermal stability of the PU film.

Based on the above analysis, we could explain that the improvement of the comprehensive properties of the modified PU should be ascribed to the interfacial compatibility of the PU and nanocellulose, in addition to the excellent nature of the nanocellulose.

## 3. Materials and Methods

### 3.1. Experimental Materials

The raw material for extracting nanocellulose is Amorpha fruticosa Linn., taken from the suburb of Tai’an, Shandong Province, China. Other required reagents include ethanol, toluene, sodium chlorite, potassium hydroxide (purchased from Tianjin Kaimi Chemical Industry Co., Ltd., Tianjin, China), 2,2,6,6-tetramethylpiperidine-1-oxide (Aladdin Biochemical Technology Co., Ltd., Shanghai, China), which are directly used without further purification.

Reagents needed to prepare waterborne polyurethane include isophorone diiso- cyanate (Jining Hongming Chemical Reagent Co., Ltd., Jining, China), polypropylene glycol 2000 (Qingdao Yousuo Chemical Co., Ltd., Qingdao, China), 1,4-butanediol, dimethylolpropionic acid, triethylamine, dibutylamine, bromocresol green (Shandong Xiya Chemical Industry Co., Ltd., Linyi, China), dibutyltin laurate (Tianjin Damao Chemical Reagent Factory, Tianjin, China), *N*-methyl pyrrolidone (Tianjin Zhiyuan Chemical Reagent Co., Ltd., Tianjin, China), distilled water (homemade), toluene, acetone, concentrated hydrochloric acid (Taian Keshang Biotechnology Co., Ltd., Taian, China), isopropanol (Tianjin Kaitong Chemical Reagent Co., Ltd., Tianjin, China). All of them are analytical grade and directly used. Closed primer, primer and contrast finish (Jiabaoli Chemical Group Co., Ltd., Jiangmen, China), are directly used.

### 3.2. Experimental Methods

#### 3.2.1. Preparation of Nanocellulose

Briefly, the nanocellulose is derived via the subsequent processes: (1) Removal of lignin and hemicellulose to derive purified cellulose fibers; (2) Carboxylation treatment by the TEMPO oxidation method; (3) High-pressure homogenization treatment to obtain the aimed nanocellulose. The detailed processes can be referred to our previously published papers [27,28,29,30,32].

#### 3.2.2. Preparation of Waterborne Polyurethane Emulsion

(1)Dehumidification treatment of Polypropylene glycol 2000, 1,4-butanediol and dimethylolpropionic acid under vacuum conditions (0.09 MPa) at 110 °C for 100 min;(2)Reaction of isopropanone diisocyanate and polypropylene glycol 2000 at a molar ratio of 1.5:1 under 65 °C for 1.5 h.(3)Chain extending process of adding a quantitative solution of 1,4-butanediol/acetone into the above solution (step 2) at 70–80 °C for 1.5 h.(4)Hydrophilic process of quantitatively adding dimethylolpropionic acid/*N*-methylpyrrolidone solution into the step 3 solution at about 70 °C for 3.5 h, during which acetone is employed to regulate the viscosity.(5)Neutralization process of adding an appropriate amount of triethylamine into the reaction system at room temperature and stirring for 15 min.(6)Emulsifying process of adding quantitative deionized water into the above solution of step 5 at high-speed blending of 12,000 rpm for 20 min to obtain an aqueous polyurethane emulsion.

#### 3.2.3. Physical Modification of Waterborne Polyurethane by Nanocellulose

Waterborne polyurethane emulsion was prepared according to the above processes (Section 3.2.2) from step 1 to step 5, followed by emulsification with same amount of nanocellulose suspension instead of the deionized water according to the step 6. Then, the mixed solution was ultrasonically treated under 1000 W for 10 min to uniformly disperse the nanocellulose. After that, the physically-modified waterborne polyurethane by nanocellulose was finally derived.

Note: three mass ratios of nanocellulose accounting for the solid content of polyurethane (0.1%, 0.2%, and 0.4%) were designed for optimization of the physical modification.

#### 3.2.4. Chemical Modification of Waterborne Polyurethane by Nanocellulose

Waterborne polyurethane emulsion was prepared according to the above processes (Section 3.2.2) from step 1 to step 6; except that in step 2, the nanocellulose after acetone replacement of water was firstly reacted with isophorone diisocyanate at 65 °C for 1 h, followed by the addition of polypropylene glycol-2000 at 65 °C for a further reaction of 1.5 h.

Note: three mass ratios of nanocellulose accounting for the solid content of polyurethane (0.1%, 0.2%, and 0.4%) were designed for optimization of the chemical modification.

#### 3.2.5. Characterization and Properties Evaluation of the PU Paint

(1)Scanning electron microscopy (SEM) observation. The microstructures of the PU films were observed by scanning electron microscopy (FE-SEM, JEM-6610LV, JEOL USA Inc., Peabody, MA, USA). The test conditions include high vacuum mode, working voltage of 12.5 kV, and beam spot of 5.0.(2)Transmission electron microscopy (TEM) observation. The derived different PU emulsions were observed by transmission electron microscope (TEM, JEM-1400, JEOL USA Inc., Peabody, MA, USA). The PU emulsion was dropped onto copper screen and then negatively stained by phosphotungstic acid and finally dried at room temperature before examination.(3)Atomic force microscopy (AFM) observation. The derived different PU emulsions were characterized by Atomic Force Microscope (AFM, NaioAFM, Nanosurf AG, Liestal, Switzerland) with tapping mode. The PU emulsion was dropped onto mica plate and dried at room temperature for further examination.(4)X-ray diffraction (XRD) characterization. The crystal structure and crystallinity of the PU films were characterized by X-ray diffractometer (XRD, D/max 2200, Rigaku Americas Corporation, Woodlands, TX, USA). The test parameters include a copper target, ray wavelength of 0.154 nm, scanning angle from 5° to 60°, scanning speed of 4 (°)/min, step of 0.02°, voltage of 40 kV, and current of 30 mA.(5)Fourier transform infrared (FTIR) characterization. The FTIR spectra were obtained using a Nicolet Magna 560 FTIR instrument (Thermo Nicolet Inc., Madison, WI, USA). The test parameters were resolution of 4 cm-1 and scans number of 32 times. Placing the sample on the diamond ATR accessory of the sample stage and adjusting the pressure column to the appropriate location for the test.(6)Thermogravimetric (TG) characterization. The thermal stability of the PU films were tested by a Thermogravimetric Analyzer (TGA Q500, Waters, New Castle, DE, USA) instrument. Five to ten mg samples were employed for the test with conditions of continuous nitrogen flow, heat rate of 10 °C/min and the temperature ranged from 35 °C to 450 °C.(7)Tensile strength and elongation at break measurement—the test was carried out using a microcomputer-controlled electronic universal testing machine (Jinan Test Group Co., Ltd., Jinan, China) with the model of WDW-5E according to the GB/T1040-1992: “Test Methods for Tensile Properties of Plastics.” The paint film was cut into dumbbell shape with standard cutter, which was 115 mm in total length, 80 mm in clamp space, 35 mm in gauge length and 6 mm in width in the middle parallel part. Each test result was mean value of three experimental data.(8)Abrasion resistance test. The test was carried out using a BGD523 type paint film abrasion tester according to the GB/T1768-2006: “Physical and Chemical Performance Test of Furniture Surface Paint Film—Part 8: Measurement Method of Abrasion Resistance.” Three pieces of maple veneer were sprayed and cut into square pieces of 100 mm × 100 mm with a small hole in the middle. The abrasion test was conducted under conditions of 800# sandpaper pasted on the grinding wheel with double-sided tape and the two arms with 1000 g weight pressed on the wood samples for rotation of 250 circles. The experimental results were averaged from three experimental data that were measured by weighing the mass loss before and after abrasion.(9)Glossiness test. The test was carried out using a GZ-II three-angle gloss tester according to GB/T 4896.6-2013: “Physical and Chemical Performance Test of Furniture Surface Paint Film—Part 6: Gloss measurement.” Three pieces of Maple veneer were sprayed with the PU paint and the glossiness was the mean value of three experimental data. Before the test, the gloss tester was calibrated by a standard plate.(10)Hardness test. The test was conducted by a pendulum hardness tester (Guangzhou Biuged BGD 508) according to GB/T 1730-1993: “Determination of Paint Film Hardness—–Pendulum Damping Test.” The result was the mean value of three experimental data.(11)Drying time test—the test was carried out according to GB/T 1728-1979: “Method for determining drying time of paint film and putty film.” The drying process was carried out in a constant temperature and humidity chamber with temperature of 30 °C and humidity of 50%. The experimental result was the mean value of three experimental data.

## 4. Conclusions

From the above analysis, we conclude as follows:(1)Nanocellulose, with a 10nm diameter and an aspect ratio of over 1000—which is derived from a biomass material by the TEMPO oxidation method—could be uniformly dispersed in the waterborne polyurethane emulsion with an entangled network structure.(2)The tensile strength, tensile elongation at break, glossiness and hardness of the CM PU and the PM PU at 0.1 wt% nanocellulose addition presents the highest value in the three nanocellulose additions of the corresponded PU modification, respectively; and the CM PU reaches the highest values when compared to the PM PU and the control PU; suggesting that such a method could effectively improve the comprehensive properties of PU and broaden the applications of nanocellulose and waterborne PU coating.

## Figures and Tables

**Figure 1 molecules-24-03151-f001:**
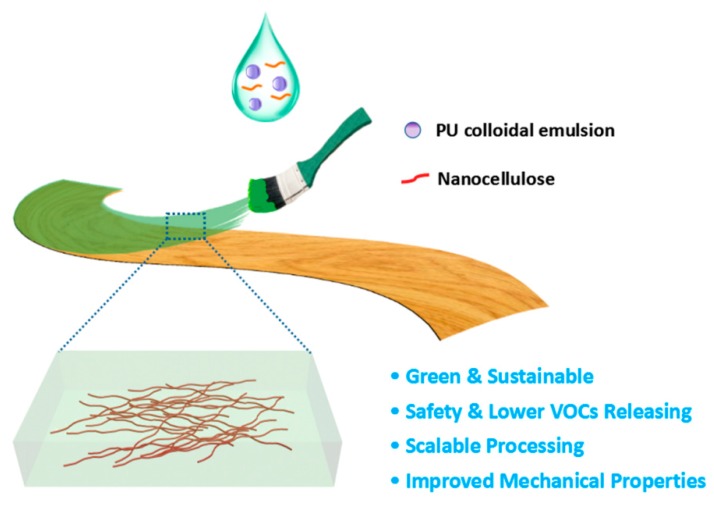
Schematic illustrations of nanocellulose-reinforced waterborne polyurethane as wood coating.

**Figure 2 molecules-24-03151-f002:**
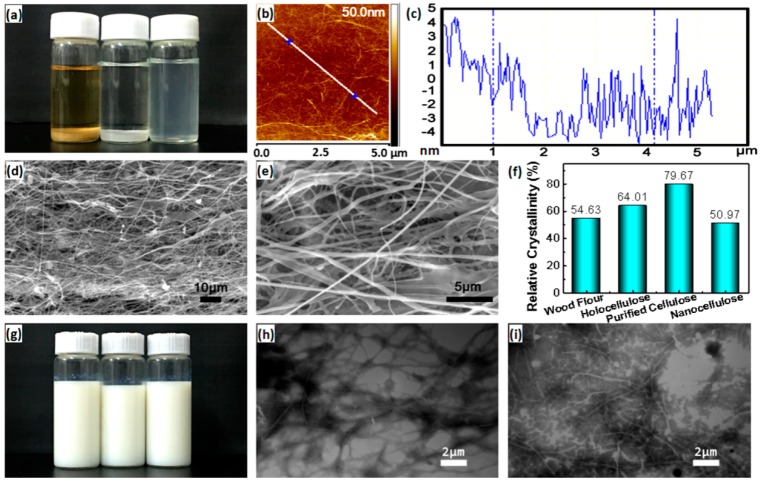
Nanocellulose morphology and structural characterization. (**a**) Photograph of the evolution of raw material morphology during the preparation of nanocellulose, from left to right: raw wood flour, purified cellulose and nanocellulose in water; (**b**) Atomic force microscopy (AFM) characterization of nanocellulose; (**c**) Diameter distribution of nanocellulose corresponding to the (**b**); (**d**) Scanning electron microscopy (SEM) photograph of nanocellulose; (**e**) An enlarged SEM photograph of (**d**); (**f**) Relative crystallinity change during the preparation of nanocellulose; (**g**) Photograph of different waterborne polyurethane (PU) emulsions (left, control PU emulsion; the middle one contains 0.1 wt% of nanocellulose by physical addition; the right one contains 0.1 wt% of nanocellulose by chemical addition); (**h**) Transmission electron microscopy (TEM) photograph of waterborne polyurethane emulsion with 0.1 wt% nanocellulose addition by physical blending way; (**i**) TEM photograph of waterborne polyurethane emulsion with 0.1 wt% nanocellulose addition by chemical modification way.

**Figure 3 molecules-24-03151-f003:**
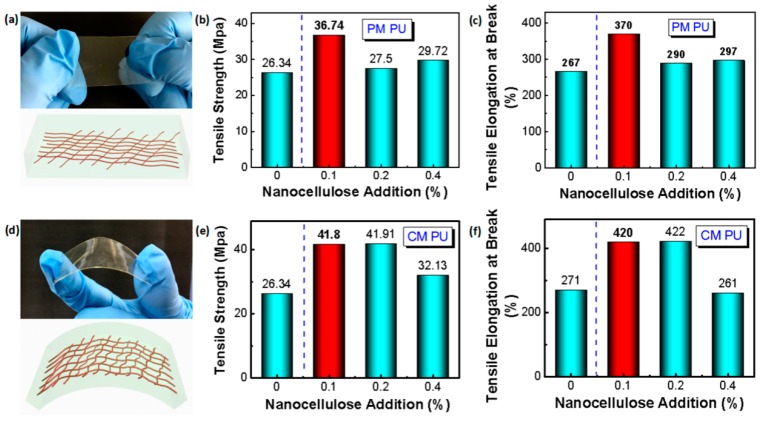
Comparison of tensile and fracture properties of the PU film. (**a**) Photograph and schematic of tensile state of the PU film physically modified by nanocellulose; (**b**) Comparison of tensile strength of physically modified PU (PM PU) with different amounts of nanocellulose addition; (**c**) Comparison of elongation at break of physically modified PU with different amounts of nanocellulose addition; (**d**) Photograph and schematic of the bending state of the PU film chemically modified by nanocellulose; (**e**) Comparison of tensile strength of chemically modified PU (CM PU) with different amounts of nanocellulose addition; (**f**) Comparison of elongation at break of chemically modified PU with different amounts of nanocellulose addition.

**Figure 4 molecules-24-03151-f004:**
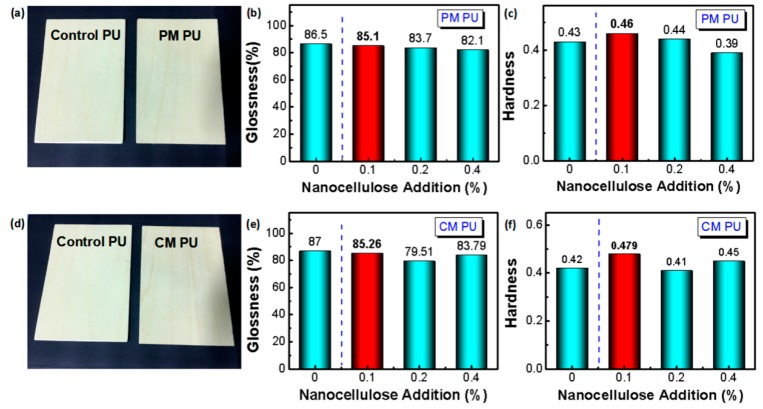
Comparison of glossiness and hardness of the PU film as wood coating with/without nanocellulose addition. (**a**) Comparison of the appearances of the control and the PM PU wood coating with 0.1wt% nanocellulose addition; (**b**) Comparison of the glossiness of the PM PU films with different nanocellulose additions; (**c**) Comparison of hardness of the PM PU films with different cellulose contents; (**d**) Comparison of the appearance of the control and the CM PU wood coating with 0.1wt% nanocellulose addition; (**e**) Comparison of the glossiness of the CM PU films with with different nanocellulose contents; (**f**) Comparison of hardness of the CM PU films with different nanocellulose contents.

**Figure 5 molecules-24-03151-f005:**
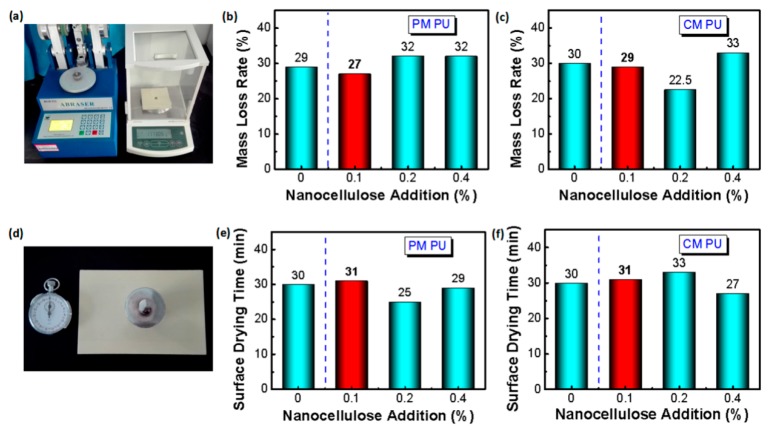
Comparison of abrasion resistance and surface drying time of the PU coatings. (**a**) Measurement of abrasion resistance of the PU coatings. (left: abrasion apparatus, right: electronic balance); (**b**) Comparison of abrasion resistance of PM PU coatings with different amounts of nanocellulose addition; (**c**) Comparison of abrasion resistance of CM PU coatings with different amounts of nanocellulose addition; (**d**) Measurement of surface drying time of the PU coatings. (left: clock, right: weight on the wood coating); (**e**) Comparison of surface drying time of PM PU coating with different amounts of nanocellulose addition; (**f**) Comparison of surface drying time of CM PU coating with different amounts of nanocellulose addition.

**Figure 6 molecules-24-03151-f006:**
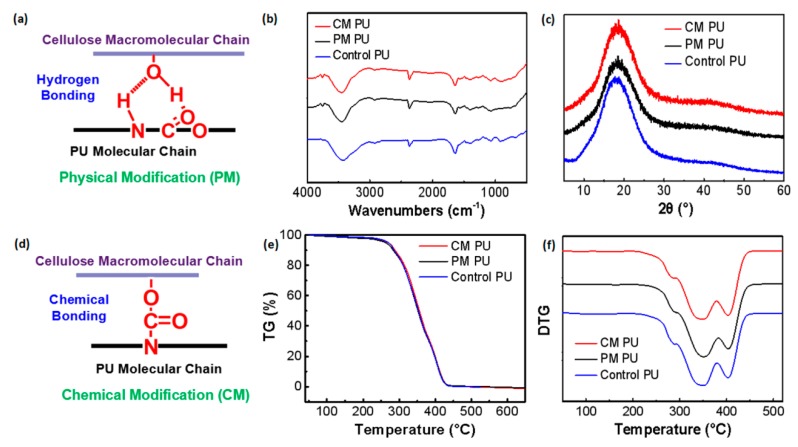
Comparison of the compositions and thermal stabilities of the PU coatings with/without modification. (**a**) Schematic diagram of the hydrogen bond formed between nanocellulose (0.1wt%) and the PU macromolecular chains by physical blending method; (**b**) Comparison of the FTIR curves of the two modified PU films (0.1 wt% nanocellulose addition) and the control PU film; (**c**) Comparison of the X-ray diffraction (XRD) curves of the two modified paint films (0.1 wt% nanocellulose addition) and the control PU film; (**d**) Schematic diagram of the chemical bond formed between nanocellulose and the PU molecular chains under chemical modification method; (**e**) Comparison of the thermogravimetric (TG) curves of the two modified paint films (0.1 wt% nanocellulose addition) and the control PU film; (**f**) Comparison of the DTG curves of the two modified paint films (0.1 wt% nanocellulose addition) and the control paint film.

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
