# Peer review of "Nanocellulose-Reinforced Polyurethane for Waterborne Wood Coating"

_molecules, 2019, doi:10.3390/molecules24173151_

Round 1

Reviewer 1 Report

Waterborne Polyurethane Emulsions are coatings and adhesives that use water as the primary solvent. With increasing federal regulation on the amount of volatile organic compounds (VOCs) and hazardous air pollutants (HAPs) that can be emitted into the atmosphere, Polyurethane Emulsions are being used in many industrial and commercial applications. Waterborne Polyurethane Emulsions are highlighted for their ability to provide good film hardness, varying degrees of flexibility, durability, solvent resistance, chemical resistance and water resistance. By using modified PU, we are able to obtain coatings properties that would not be possible if other types of resins were used. Formulators can achieve different coatings properties by varying the type of PU modification chosen.

The paper reviewed reported about the nanocellulose-reinforced PU for Waterborne Wood coating.

General Comments:

1.    The authors don´t follow the general template recommended for the Molecules journal, please revise and applied them (https://www.mdpi.com/journal/molecules/instructions).

2.    The English is generally readable, but your submission will get better peer review if you have the text edited before submission. Alternatively, students in my lab for whom English is not a first language swear by Grammarly.

3.    Typing errors must be reviewed (i.e. : line 4. # as superscript for the three first authors; or the general incorrect use of um for micrometers in place of mm).

4.    Adhesion test, water resistance test, but also alkali resistance test  are described in section 2.2.5, however, the final results are not included in any part of the manuscript. A brief explanation related to mention parameters should be included in section 3.

5.    The follow recent patents should be included as references of the present report:

-Method for preparing nanocellulose-polyurethane iridescent structural color coating for wood products. Apr 05, 2019, CN 109575696, A.

-Method for chemically modifying waterborne wood lacquer containing hemicellulose nanocellulose. Jan 22, 2019, CN 109251648, A.

- Nano-scale cellulose functional doping modified coating and preparation method thereof. Dec 21, 2018, CN 109054584, A

In summary, I recommend the publication of this paper after major revisions. Some comments and specific suggestions could be taking into account by the authors.

Specific comments follow:

Abstract

Line 17. Included Volatile Organic solvents previous of its abbreviation (VOCs).

Introduction

          Line 53. The paragraph started with the phrase “In recent years” but the most recent reference included is from 2015 (one reference), 2014 (four references), 2011 (two references), and 2010 (two references). In my opinión, Recent years must understand the last 3-5 years.

Experimental materials and methods

          The title of this section should be change by Materials and Methods

          If a Supporting Information is missing, the main results related to the characterization of modified PU should be included in this section at the end of each technique.

Experimental results and discussion

          The title of this section should be change by Discussion

          Figure 2ª, included in the corresponding food page “Photograph of the evolution of raw material morphology during the preparation of nanocellulose, from left to right: raw wood flour, purified cellulose and 214 nanocellulose in water”

          Regarding TEM analysis (Figure 2h and 2i), the authors concluded that “the nanocellulose is uniformly dispersed in the waterborne polyurethane in the network winding state”, however, the appereance of PM-PU( 2h)  and CM-PU (2i) under TEM are not the same. On the other hand, althought the authors use differents mass ratios of nanocellulose (0,1%, 0,2%, 0,4%) for its formulations, for the AFM, SEM and TEM analysis, reference is only made to 0.1%. A more complete analysis should be included.

          For he Figures 3a and 3d it is not clear if the authors showed your owns pictures, or which is the main differences betwwen both pictures, please, define.

          At least twice, the authors mention the degree of aggregation of nanocellulose to explain some results, however, there is no evidence of this in the article, which could easily be demonstrated with photos of TEM or SEM.

          Line 293: Replace “nanocllulose” by “nanocellulose”.

          I recommend standardizing the use of significant figures for all reported data, but specifically focuse on the data shown in the graphs in Figure 4.

          Page 9: A clear explanation related to Mass loss rate and Surface Drying Time should be included in place of “so on” present in the 343 line.

          Line 344, The phrase “In a word” should be replaced by another more academic phrase like “Summarizing” or “In short”.

        Line 344-352. In spite of the analysis of all parameters, the CM PU (0,1wt% of nanocellulose addition) seems to be the better choice. Does this mean that some parameters are more important than others?

          Line 369-378. In this paragraph, the FTIR analysis is carried out. However, the three FTIR curves from Figure 6b are so similar. The described results are not explicit from the corresponding figure.

          Line 382-384. Regarding the Thermal stability analysis (Figure 6e), the results showed that the addition of nanocellulose has not significative effect in this parameter. However, theoretically I would expected higher TS for the CM PU because in this case covalent bond need to be broken vs PM PU in which only H bond (intermolecular forces) are present. Why TS are so similar between CM PU, PM PU and control PU?

Conclusions

          The different ways to modified waterbourne PU emulsion, but also the advantage and/or diffrences between both, CM PU and PM PU vs PU should be included.

Reviewer 2 Report

Thank you all authors for your submitted article. Please consider following items:

the number indicated for wood, hemicellulose and extracted cellulose crystallinity is not in agreement with published fact. How did you measured crystallinity. it is necessary to indicate x-ray spectra. 

it is necessary to indicate stress/strain graph on top of just results. 

Hard to detect the signals mentioned in the article about  FTIR spectra. Need better graph

Round 2

Reviewer 1 Report

No more comments or suggestions.

I accept the paper Nanocellulose-reinforced PU for Waterborne Wood Coating in the present form.

Best regards